

# Habitat preference, movements and growth of giant mottled eels, *Anguilla marmorata*, in a small subtropical Amami-Oshima Island river

Hikaru Itakura[1,2,*] and Ryoshiro Wakiya[3,4,*]

[1] Chesapeake Biological Laboratory, University of Maryland Center for Environmental Science, Solomons, MD, United States of America
[2] Graduate School of Science, Kobe University, Kobe, Hyogo, Japan
[3] Research and Development Initiative, Chuo University, Bunkyo-ku, Tokyo, Japan
[4] Atmosphere and Ocean Research Institute, University of Tokyo, Kashiwa, Chiba, Japan
[*] These authors contributed equally to this work.

## ABSTRACT

Although anguillid eel populations have decreased remarkably in recent decades, few detailed ecological studies have been conducted on tropical eels such as the giant mottled eel whose range extends across the whole Indo-Pacific. This species was studied throughout the entire 0.5 km mainstem reaches of Oganeku River on the subtropical Amami-Oshima Island of Japan over a two-year period using four sampling periods to understand its habitat preference, early life-stage dispersal process, movements, and annual growth using a mark-recapture experiment conducted with quantitative electrofishing. A total of 396 juvenile growth-phase *A. marmorata* eels were caught and tagged, with 48 individuals being recaptured at least once. Their density irrespective of size of eels was most strongly determined by distance from the river mouth, followed by riverbank type according to random forest models. Eel density decreased with increasing distance from the freshwater tidal limit located about 100–150 m from the river mouth. Eels preferred vegetated riverbanks, while they avoided those of concrete and sand. The density of small eels (total length: TL < 240 mm) was also associated with depth and velocity, with small eels tending to prefer riffle or run habitats. In contrast, large eels (TL ≥ 240 mm) were found in habitats of any depth and velocity. The TL of eels had a minimum peak at around the tidal limit, and it increased with increasing distance from the tidal limit. The observed density and size gradients of eels in relation to the distance from the river mouth suggested that *A. marmorata* initially recruited to freshwater tidal limit areas and then dispersed in both downstream and upstream directions. The growth rate of eels varied greatly among individuals that were at large for various periods of time and ranged from 0 to 163.2 mm/year (mean ± SD of 31.8 ± 31.0 mm/year). Of the recaptured eels, 52.1% were recaptured in a section that was different from the original capture section, and their mean ± SD distance travelled was 46.5 ± 72.5 m (median = 20 m). 47.9% of the eels were recaptured from the original section of capture (i.e., <10 m distances travelled), suggesting that they had strong fidelity to specific habitats with limited movements. The distance travelled of eels that had moved was greater for small eels (range = 10–380 m; mean ± SD = 84.4 ± 121.9 m) than large individuals (range = 10–120 m; mean ± SD = 30.9 ± 31.0 m), which indicates that the mobility

Corresponding author
Hikaru Itakura,
itakurahikaru@gmail.com

of the eels declines as they grow. This is the first clear detailed documentation of the spatial distribution, growth, and movements of tropical eels in a small river system in relation to environmental conditions that provides an example of how future studies can be conducted in other areas to understand how conservation efforts can be most efficiently targeted for maximum success.

## INTRODUCTION

The eels of the genus *Anguilla* comprise 16 species of catadromous fishes that undertake large-scale oceanic migrations between their offshore spawning areas and growth habitats in continental waters during their life histories. Populations of anguillid eels are distributed throughout much of the world from tropical to temperate regions that include more than 150 countries (*Jacoby et al., 2015*). Ten species are distributed in tropical regions (i.e., tropical eels), whereas the remaining six species are distributed in temperate regions (i.e., temperate eels). Anguillid eels have ecological, commercial, and cultural importance in many regions (*Jacoby et al., 2015*) and are increasingly considered as important representative species for freshwater biodiversity conservation efforts (*Itakura et al., 2020a*). Because of population declines, 10 of the 16 anguillid eel species (3 of which are subdivided into subspecies) are listed as ''Threatened'' or ''Nearly Threatened'' in the International Union for Conservation of Nature (IUCN) Red List of Threatened Species (*IUCN, 2019*). Although, the northern hemisphere temperate anguillids have shown well-documented declines, some tropical eels are also of concern for conservation even though their freshwater ecology has not been studied and they are of lower economic commercial importance than the more extensively studied temperate species (*Jacoby et al., 2015*). Thus, ecological knowledge about tropical eels is clearly essential for the conservation and management of anguillid eels in the Indo-Pacific.

One of the tropical eels, the giant mottled eel, *A. marmorata*, is the most widespread anguillid species in the world, because it is distributed in the Indian Ocean, and across the Indo-Pacific to French Polynesia in the South Pacific Ocean (*Ege, 1939*; *Watanabe, Aoyama & Tsukamoto, 2004*). The species has multiple genetically distinct populations (*Minegishi, Aoyama & Tsukamoto, 2008*), one of which spawns in the North Equatorial Current region of the western North Pacific Ocean, where the Japanese eel, *A. japonica*, spawns (*Kuroki et al., 2009*; *Tsukamoto et al., 2011*). Because of a recent increase in demand for *A. marmorata* as a fisheries/aquaculture replacement for temperate eels, especially in East Asia (*Gollock et al., 2018*), understanding the ecology of *A. marmorata* is particularly important.

Ecological aspects of growth-phase anguillid eels have been extensively studied in temperate eels such as the American eels (*A. rostrata*), European eels (*A. anguilla*), and *A. japonica*. After larval development and migration in the open ocean, the leptocephalus

larvae of anguillid eels metamorphose into glass eels (early juvenile phase) that enter rivers. Glass eels appear to initially accumulate at the freshwater tidal limit of estuaries, and then disperse in both upstream and downstream directions (*Haro & Krueger, 1991*; *Edeline et al., 2007*; *Kaifu et al., 2010*; *Wakiya et al., 2019*). They then settle and spend their growth phase in a variety of habitats ranging from saline bays or brackish estuaries to rivers all the way to upland headwaters, and they also live in lakes (*Moriarty, 2003*). While small eels exhibit dispersal or movement behaviors (*Laffaille, Acou & Guillouët, 2005*; *Imbert et al., 2010*), large individuals mostly display sedentary behavior with limited movements and small home ranges (*Gunning & Shoop, 1962*; *Parker, 1995*; *Jellyman & Sykes, 2003*; *Ovidio et al., 2013*; *Itakura et al., 2018*). Some individuals however, are increasingly realized to exhibit habitat shifts or seasonal movements between the different habitats (*Jessop et al., 2002*; *Daverat et al., 2006*; *Yokouchi et al., 2012*; *Béguer-Pon et al., 2015*).

Many studies have reported that the abundance of eels in rivers decline with increasing distance from the tidal limit of estuaries (*Ibbotson et al., 2002*; *Aprahamian et al., 2007*; *Costa et al., 2008*; *Kaifu et al., 2010*; *Wakiya et al., 2019*). Riverine distribution of eels can be also affected by other environmental factors in microhabitats such as depth, velocity, sediment, aquatic vegetation, riverbank conditions, effects of which can differ depending on the body size of eels (*Glova, Jellyman & Bonnett, 1998*; *Laffaille et al., 2003*; *Kume et al., 2019*). These types of environmental factors have the potential to influence the distribution of *A. marmorata*.

The basic habitat use patterns of tropical eels has been studied recently in a few locations where qualitative sampling or otolith microelement analysis were used. Several studies have found that although *A. marmorata* tends to live in freshwater areas rather than in brackish and marine habitats (*Shiao et al., 2003*; *Nguyen, Tsukamoto & Lokman, 2018*; *Hsu, Chen & Han, 2019*), the species can occupy a broad range of habitats from brackish estuaries to upland headwaters (*Arai & Chino, 2018*; *Hagihara et al., 2018a*; *Wakiya, Itakura & Kaifu, 2019*; *Kumai, Tsukamoto & Kuroki, 2020*). There are often sympatries of multiple eel species in tropical rivers that appear to affect the patterns of habitat use among the species presumably to reduce interspecific competition (*Marquet & Galzin, 1991*; *Arai & Abdul Kadir, 2017*; *Hagihara et al., 2018a*), and sympatries of temperate and tropical eels also occur in subtropical regions of their distribution ranges (*Shiao et al., 2003*; *Hsu, Chen & Han, 2019*; *Itakura et al., 2020b*). In rivers where a single eel species such as *A. marmorata* is highly dominant among anguillid species, it is found throughout the river network (*Robinet et al., 2007*; *Itakura et al., 2019*; *Wakiya, Itakura & Kaifu, 2019*).

The growth rate (GR) of anguillid eels is another key aspect of their ecology that reflects many characteristics of the environments where they live. It has been intensively studied for temperate eels (e.g., *Vøllestad, 1992*; *Morrison & Secor, 2003*; *Daverat & Tomás, 2006*; *Yokouchi et al., 2008*), and GR was recently studied for tropical eels as well (*Hagihara et al., 2018b*; *Wakiya, Itakura & Kaifu, 2019*; *Kumai, Tsukamoto & Kuroki, 2020*). There is considerable intra-interspecific variation in the annual GR of eels that is related to their latitudinally expanded distributional regions (*Hagihara et al., 2018b*) and the different environments of the wide-range of continental habitats where eels are present (*Morrison & Secor, 2003*; *Yokouchi et al., 2008*). Eel GR also varies substantially among different

ages, years, and individuals (*Yokouchi & Daverat, 2013*). The annual GR of eels has usually been calculated by dividing the body length at capture by age that is estimated based on otolith annual rings after the eels recruit as glass eels, but this method probably overlooks extremely low or high GRs due to inconsistent ring deposition, which would not reflect the actual diversity the GR of eels.

Therefore, ideally, the actual increase of body length during a known period of time of individual eels should be directly measured by mark-recapture experiments, which will provide a more precise understanding of their growth strategies. Our objective was to conduct a comprehensive survey and mark-recapture experiment for two years in a small subtropical island river to understand the habitat preference, early life-stage dispersal, movement, and growth of giant mottled eels on Amami-Oshima Island, Japan. *Anguilla marmorata* is clearly the dominant anguillid species throughout the rivers in this island (*Wakiya, Itakura & Kaifu, 2019*; *Itakura et al., 2020b*; *Itakura et al., 2020a*), thus this island offers suitable study sites for a case study to investigate their ecology in small rivers that have minimal interspecific competition.

## MATERIALS AND METHODS

### Study area and sampling

This study was conducted in the Oganeku River on Amami-Oshima Island, Kagoshima Prefecture, Japan (Fig. 1; 28°21′42.2″N 129°21′03.4″E). Amami-Oshima Island is located between the southern mainland of Japan and Okinawa Island adjacent to the western North Pacific Ocean and next to the Kuroshio Current that is one of the strongest western boundary currents. This is the second largest island in the Nansei Islands (Okinawa is the largest) in terms of area (712.35 km$^2$). The climate of this island is characterized by a warm and wet climate with an average annual temperature of 21.6 °C (monthly range: 14.8–28.7 °C), with a peak in July and annual precipitation of 2837.7 mm (monthly range: 156.9–410.3 mm) with a peak in June (1981–2010 data of the Japan Meteorological Agency, https://www.jma.go.jp/jma/index.html).

The study river is approximately 0.5 km in length, and flows through agricultural and forest lands (Fig. 2). The elevation of the river increases dramatically to >10 m from around the 320 m area from the river mouth where the riverscape transitioned to higher-gradient upstream environments. There is a waterfall >10 m height at 520 m from the river mouth (elevation = 50 m). Although some eels might be able to climb the waterfall, we surveyed from the river mouth to the waterfall, because we were not able to access areas above the waterfall. Because the river width around the waterfall is much narrower (<1 m), the river flow may originate not far above the waterfall. Thus, our surveyed area was considered to cover almost all of the mainstem of the river. The width of the river was 3.2 ± 2.0 m (mean ± SD; range: 0.5–8.4 m), and the depth was 22.2 ± 16.7 cm (range: 3–75 cm). The freshwater tidal limit of the river was observed during our surveys to be located at about 100–150 m from the river mouth based on tidally influenced increases in water depth during high tides. The freshwater areas of this island are dominated by diadromous species (*Itakura et al., 2020a*). A total of 33 species (24 fishes and 9 crustaceans) was identified

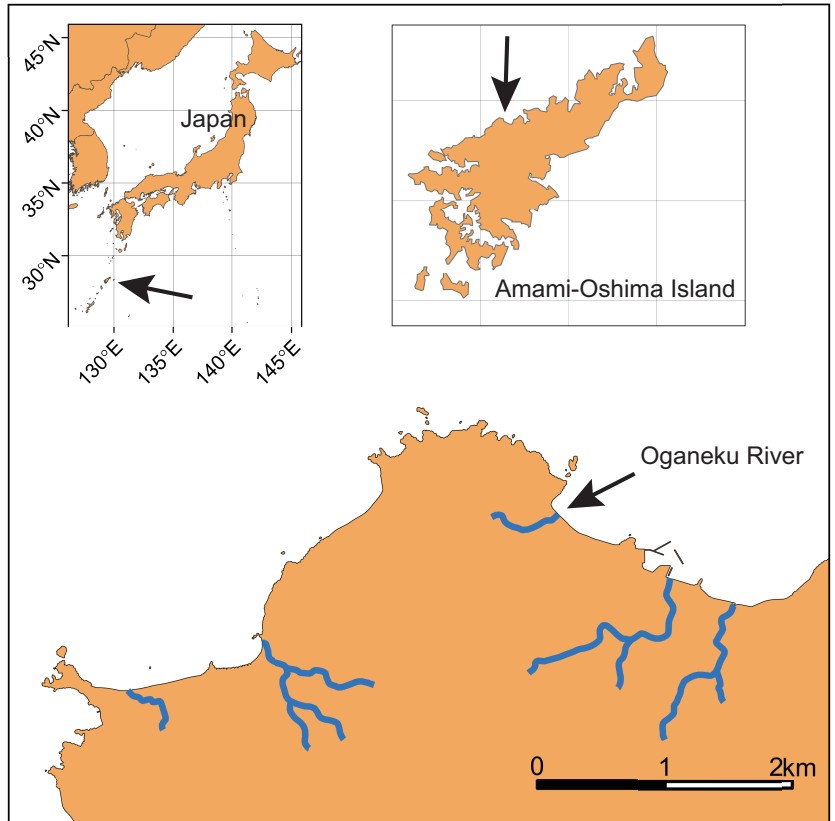

**Figure 1 Locations of Amami-Oshima Island, Japan and the Oganeku River.** The study on giant mottled eels *Anguilla marmorata* was conducted during four sampling periods throughout the entire river drainage.

in the study river during the sampling surveys, all of which were diadromous species (Table S1). *A. marmorata* was the dominant fish species in terms of both abundance and biomass in the river. This island is near the northern limit of the distribution range of *A. marmorata* (*Jacoby & Gollock, 2014*), but some eels also recruit to areas farther north in mainland Japan (*Mizuno & Nagasawa, 2010*).

We chose this river because (1) such a small stream allowed us to conduct quantitative sampling throughout all the main reaches of a river using electrofishing, and (2) there are no artificial migration barriers (e.g., weirs and dams) that can impede eel movement in the river, thus providing a good model system to examine their ecology without the effects of barriers. A recent study conducted in other rivers on this island showed that the density of *A. marmorata* was strongly negatively associated with cumulative height of the barriers (*Itakura et al., 2020a*).

Quantitative sampling for eels was carried out a total of four times during August and November 2016, July 2017, and September 2018 (25 month period). We captured anguillid eels over almost the entire area of the river from the river mouth to the uppermost reaches of the river (below the waterfall) using a back-pack electroshocker (LR-20B, Smith-Root, Inc.,

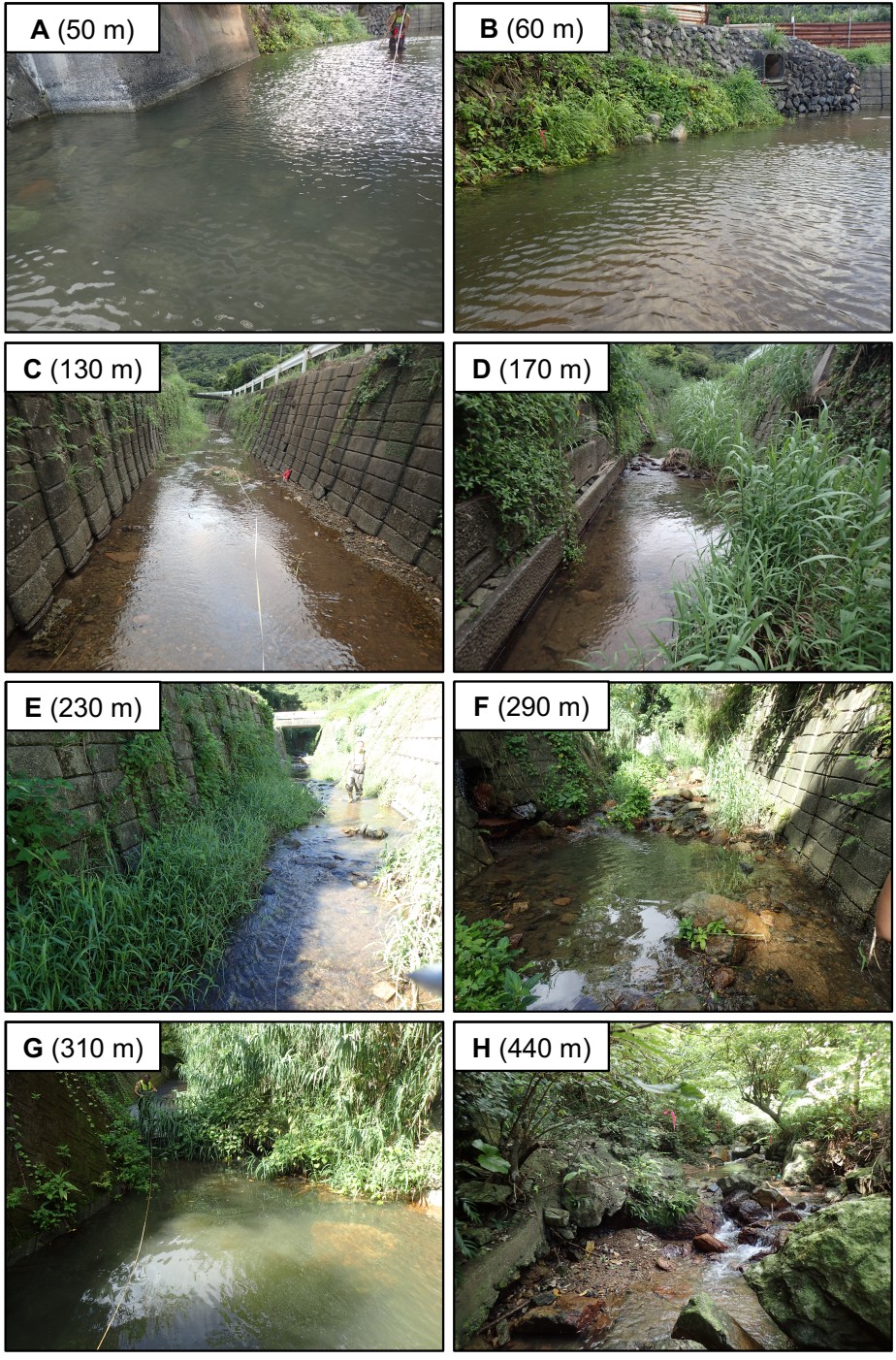

**Figure 2** **Photographs of representative study sections of the Oganeku River.** (A) 50 m, (B) 60 m, (C) 130 m, (D) 170 m, (E) 230 m, (F) 290 m, (G) 310 m, (H) 440 m sections from the river mouth. The greatest number of large eels ($n = 19$) was collected at boulder riverbank habitat in the section 60 m from the river mouth. Eels burrowed in boulder and vegetated riverbanks in the photos with some eels also burrowing in gaps in concrete riverbank. Moreover, large eels appeared to burrow under boulder or rocks, while small eels seemed to burrow in gravel.

Vancouver, WA, USA) and hand nets by two people during daytime. Each 10 m interval of the river channel was regarded as a sampling section (defined by the downstream border), and each section was defined by the distance from the river month. Each section was sampled by moving side to side starting from the downstream direction moving upstream, with each section being marked with wooden stakes along the riverbank. Sampling was performed during low tides at sections below the freshwater tidal limit of the river in order to avoid the effect of salinity on the efficiency of electrofishing. As a result, we confirmed the sections were freshwater during low tides. Captured eels were held in river water until they were anaesthetized with 10% eugenol solution (FA100; DS Pharma Animal Health Co., Ltd., Japan). Each specimen was identified morphologically following *Watanabe, Aoyama & Tsukamoto (2004)*, and their growth stage was confirmed based on the color of its body and pectoral fins following previous studies (*Okamura et al., 2007*; *Hagihara et al., 2012*). One sexually-maturing *A. marmorata* (580 mm in TL) was collected at the 290 m section on September 2018 and this was excluded from this study, because it might have already started its early migration to the ocean to spawn.

The TL and body weight (BW) of each eel were measured to the nearest 1 mm and 0.1 g, respectively, and then they were individually tagged and released back into the capture sites after they fully recovered from the anaesthesia. Each eel was tagged using different methods depending on its body weight. Large eels with BW ≥ 10 g were individually tagged by injecting a passive integrated transponder (PIT tag; BIO8.B03, Biomark, ID, USA; 1.4 mm diameter) tags into the abdominal cavity (228 eels), while small ones with BW < 10 g were tagged using injected visible implant elastomer (VIE tag; Northwest Marine technology, WA, USA) tags (105 eels). The small eels were individually distinguished based on a combination of different elastomer colors and the area of the body where they were injected (i.e., jaw, eye, and near anus). The sampling was conducted under the guidance and with the permission of the Fisheries Adjustment Rules of Kagoshima Prefecture (approval number: 2006-5 for 2016 and 2006-10 for 2017 and 2018).

Environmental conditions at each sampling section were measured and recorded immediately after the sampling in August 2016, which was during typical water flow conditions compared to the other sampling periods. The depth and water velocity were measured at the center of the river in the middle of each 10 m section, while the river width was measured at the downstream border of each section. The sediment was categorised into six types: mud, sand, gravel, boulder, concrete or bedrock, and mud and boulder. The riverbank was categorised into seven types by the combination of left and right banks: sand, boulder, vegetation, concrete, concrete and gravel, concrete and vegetation, and concrete and boulder (i.e., 2 classifications per section).

## Growth and movement

The growth rate (GR; mm/year) of each recaptured eel was calculated as:
$GR = ((TL_2 - TL_1)/(t_2 - t_1)) \times 365$ where $TL_1$ and $TL_2$ are TL of eels at $t_1$ (date at capture) and $t_2$ (date at recapture).

The distance travelled (m) of recaptured eels was calculated as distance between the capture and recapture sections. As we did not document where eels were captured within

each section, eel movement was quantified only when an eel was recaptured in a section that is different from the original capture section. The eel movement distance was regarded as 0 m (absence of movement, i.e., travel distance < 10 m) when they were recaptured in the same section where they were originally captured, whereas it was regarded as a 10 m movement when they were recaptured in an adjacent section. Technically, this means that the eel movements between adjacent sections could have ranged from 0 (on the section borderline) to 20 m (on opposite borderlines). Thus, for analyzing relationships between eel TL and presence of movement, we used both 10 and 20 m distances for adjacent-section movements.

## Data analysis

All statistical analyses were performed with R 3.6.0. To evaluate the riverine distribution of TL of eels, we used a generalized additive model (GAM; *gam* function in the *mgcv*) (*Wood, 2019*), which included TL as a response variable (gaussian distribution with an identity-link function), and distance from the river mouth as a predictor variable. To assess how the eel movement changed as the eels grew, we used a generalized linear model (GLM), which included either the presence or absence of eel movement (i.e., 1 or 0) as a response variable (binomial distribution with a logit-link function), and TL as a predictor variable. The relationship between TL at initial capture before each recapture and distance travelled of recaptured eels was also evaluated using a GLM with a negative binomial distribution and a log-link function. In addition, the effects of TL class (small and large eels), TL at initial capture before each recapture, the study period (i.e., duration between capture and recapture), the eel movement (i.e., distance travelled of 0 m, <80 m, ≥80 m), and the sampling section from the river mouth on the GR of recaptured eels were assessed using a GLM with a gaussian distribution and an identity-link function. The GR was log-transformed by adding 1, to meet the assumption of normality of the residuals. In the growth model, variable selections were performed according to Akaike's information criterion (AIC) using *dredge* in the package *MuMIn* (*Bartoń, 2019*). Moreover, the proportion of eels that moved and the distance travelled by eels were compared between small and large eels using the Fisher's exact test and the Exact Wilcoxon-Mann–Whitney test, respectively. TL of eels at initial capture before each recapture was also compared between eels that moved and those that did not move using the Exact Wilcoxon-Mann–Whitney test. We defined eels < 240 mm TL as small eels, and eels ≥ 240 mm TL as large eels, following previous studies for growth-phase European and Japanese eels that reported that the mobility of eels can change at around 240 mm TL (*Imbert et al., 2010*; *Wakiya, Kaifu & Mochioka, 2016*), because there is no information on the TL-mobility relationship for *A. marmorata*.

To evaluate the effects of environmental factors on the density of eels, we used the permutation-based random forest (RF) machine learning algorithm (*Hapfelmeier & Ulm, 2013*). The RF is an ensemble learning algorithm that builds a predictive regression model (forests) by taking an average from outputs of a large number of decision tree models (*Breiman, 2001*). We selected the RF algorithm, because RF (1) does not require normality or independence of the variables, (2) is able to handle non-linear relationships well, (3)

is not prone to overfitting by averaging a large number of decision tree models (*Breiman, 2001*), (4) fairly evaluates the relative importance between continuous and categorical variables without bias (*Strobl et al., 2008*), and (5) can perform variable selection and assess the relative importance among highly correlated variables (*Nicodemus et al., 2010*; *Bergmann et al., 2017*).

The density of *A. marmorata* in each sampling section was calculated by dividing the number of captured eels by area of the study section ($m^2$). The densities of eels of three size classes were used as response variables: all eels, small eels (TL < 240 mm), and large eels (TL $\geq$ 240 mm). The environmental factors including depth, water velocity, distance from the river mouth, sediment, and riverbank were used as predictor variables. We used the RF algorithm for performing multiple regressions for variable selection (*Hapfelmeier & Ulm, 2013*; *Ryo et al., 2018*). The RF algorithm with variable selection by *Hapfelmeier & Ulm (2013)* first performs a multiple regression using all predictor variables to estimate a statistical significance for each variable. Then, the RF algorithm performs a multiple regression using only significant variables to construct the final RF model and to estimate a relative importance score for each variable (see *Hapfelmeier & Ulm, 2013*; *Ryo et al., 2018* for more detail about the RF algorithm). We set the significance level to 0.01 with Bonferroni correction for the number of predictor variables following *Ryo et al. (2018)*. The relative importance score of each variable was quantified by evaluating how much model accuracy can decrease when the model removes the focal variable (*Breiman, 2001*). The modelled relationships between the predictor variables and each response variable were visualized using partial dependence plots, which represent the marginal effect of a particular response variable on the modelled function after marginalizing out the effects of all the other variables. The procedure calculates a partial dependence score that indicates the relative extent of the response variable. In our case, the higher the score, the higher the density of eels. Model performance was evaluated based on explanatory and predictive powers ($R^2$). Explanatory power was evaluated based on the coefficient of determination by comparing observed and fitted values as explained variance. In contrast, prediction power (validation accuracy) is a metric to estimate an expected model performance for prediction when a new dataset is analyzed. Prediction power was also evaluated based on the coefficient of determination using 1/3 of the samples that were not used in the tree construction, following the out-of-bag technique (*Breiman, 1996*).

We used the R script available in *Ryo et al. (2018)*, which was modified from the script by *Hapfelmeier & Ulm (2013)*. The script is based on *ctree* and *cforest* functions in the package *party* (*Strobl, Hothorn & Zeileis, 2009*) for RF modeling, *cforeststats* and *postResample* functions in the package *caret* (*Kuhn et al., 2020*) for evaluating model performance, and the *generatePartialDependenceData* function in the package *mlr* (*Bischl et al., 2020*) for partial dependence plots. All parameters in the functions were set to defaults.

## RESULTS

### Number of collected eels, size and density

A total of 396 growth-phase *A. marmorata* were collected in this study (this includes number of recaptured eels). Eels were collected in each of the 4 sampling times of August
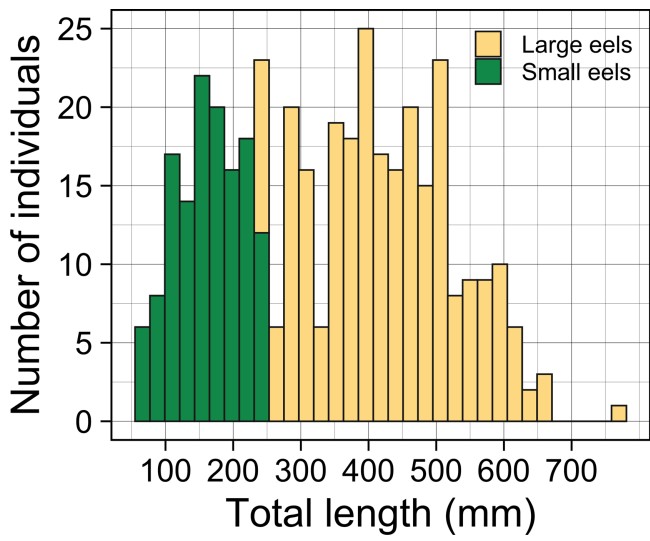

**Figure 3  Total lengths (TL) frequency histogram of *A. marmorata* eels collected in the Oganeku River.** TL of *A. marmorata* was separated into the two general size groups of large and small eels.

2016 (152 eels), November 2016 (80 eels), July 2017 (60 eels), and September 2018 (104 eels). The size of all captured eels ranged from 62 to 770 mm TL with a mean ± SD of 336.8 ± 153.0 mm (Fig. 3) and their BW ranged from 0.5 to 1219.0 g with a mean ± SD of 136.7 ± 167.5 g. The number of collected eels per sampling section ranged from 0 to 19 individuals, with the greatest single-section catch ($n = 19$) consisting of only large eels (TL ≥ 240 mm, 431.1 ± 89.4 mm, 246–590 mm) that were collected at a section 60 m from the river mouth that consisted of concrete and boulder riverbanks and mud sediment (Fig. 4). The density of eels in each section (when more than 1 individual was collected) ranged from 0.01 to 0.69 eels m$^{-2}$, with a mean ± SD of 0.15 ± 0.13 eels m$^{-2}$ (Fig. S1A). Of the collected eels, 48 individuals were recaptured at least once (39 PIT tagged eels, 9 elastomer tagged; 8 recaptures 2–3 times), and thus a total of 339 unique individuals were collected in this study. We obtained 57 records on movement events and 57 records for annual GR of the 48 recaptured eels. The TL of these eels at first-capture was 381.3 ± 131.0 mm, with a range from 105 to 656 mm, and those at recapture were 408.1 ± 135.8 mm in TL, with a range from 139 to 770 mm.

A total of seven *A. japonica* were collected at 60, 80, 90, 100, 150, 170, and 280 m sections where the sediment consisted of gravel and riverbanks consisted of boulders or vegetation, and water velocities were <40 cm$^{-2}$ (Fig. 4). Their TL was 364.0 ± 161.7 mm, with a range from 126.0 to 541.0 mm (Fig. 4C). The captured *A. japonica* were omitted from the analyses of this study.

## Size distribution

Almost all small eels (TL < 240 mm) were collected in the sections that were about 100–300 m from the river mouth, and more of the smallest eels < 100 mm TL were collected in the 100–200 m sections (Fig. 4C). Conversely, large eels (TL ≥ 240 mm) were collected
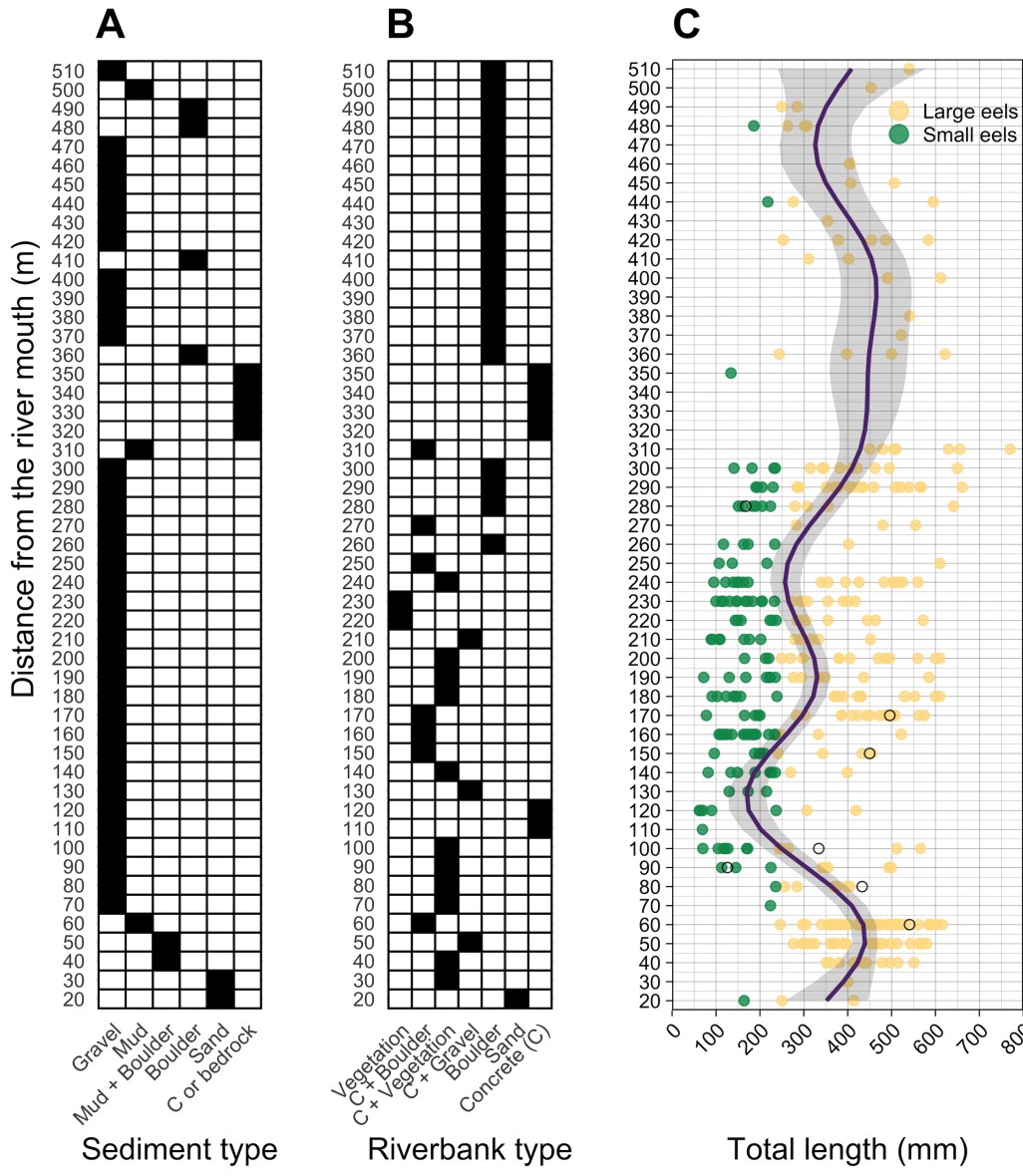

**Figure 4** **Distributions of (A) sediment, (B) riverbank types, and (C) individual Total lengths (TL) of *A. marmorata* eels in relation to distance from the river mouth in the Oganeku River.** The line and shaded area in the right panel indicate the predictive value and 95% intervals of the generalized additive model, respectively. The open circles in the right panel show the capture locations and sizes of the seven Japanese eels, *A. japonica*, that were captured during the surveys. TL of *A. marmorata* was separated into the two general size groups of large and small eels.

throughout the river (Fig. 4C), although they were not evenly distributed. Large eels were caught in 82% of the total sections, but small eels were caught only in 54% of the total sections. The GAM showed that the TL of *A. marmorata* was significantly associated with distance from the river mouth (Effective d.f. = 8.556, Reference d.f. = 8.927, $F = 15.45$, $p < 0.001$; Fig. 4C). The predicted TL reached a minimum value at around the 100–150 m sections in part due to few large eels being caught there, and it increased with increasing

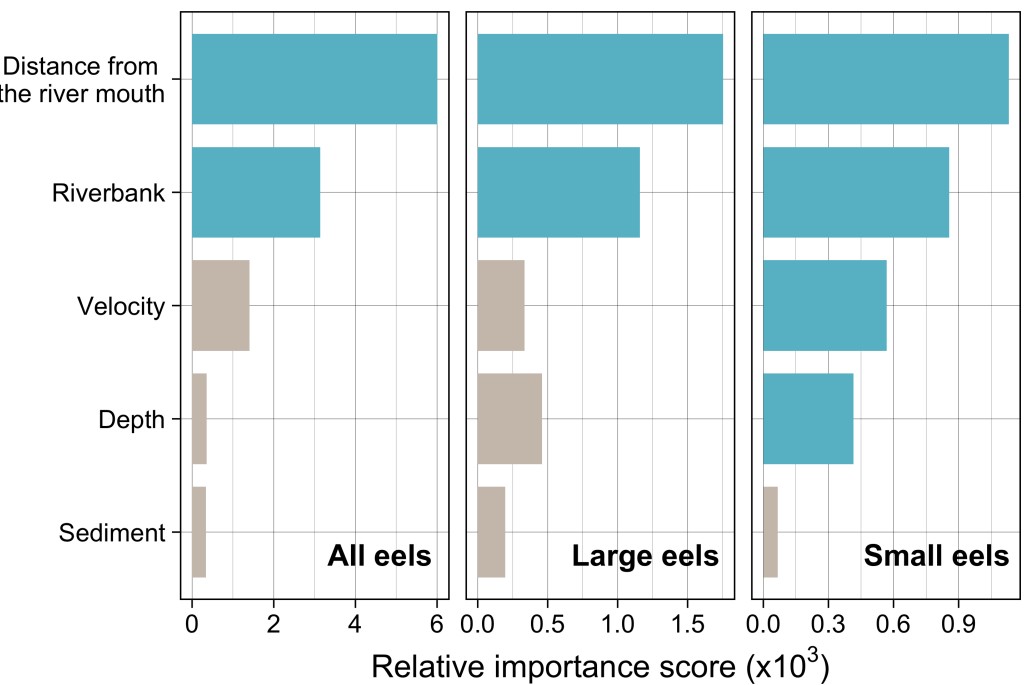

**Figure 5** Relative importance scores of all predictor variables for density of *A. marmorata* in the Oganeku River. The blue and grey bars indicate finally selected or not selected by the random forest models as a significant importance variable, respectively.

distance upstream because only 3 small eels were caught above 300 m (Fig. 4C). However, large eels ≥ 240 mm TL were caught from very near the river mouth to the farthest upstream sections, with the most eels > 600 mm TL being caught near 300 m where a wide size range (141–770 mm TL) was present in an area that was pool and run habitats with gravel or mud sediments and concrete and boulder riverbanks (Fig. 4, Figs.S1A–S1C).

## Habitat preference

Distance from the river mouth was the most important explanatory variable to predict density of eels followed by riverbank type, both of which were selected by the RF models for all three size-classes (i.e., all eels, small eels, and large eels) (Fig. 5). Conversely, velocity and depth were only selected by the model for small eels. The explanatory powers ($R^2$ value) were 45.7%, 40.9%, and 32.3% of the variation in densities of all, small, and large eels, respectively (validation accuracy: 30.3%, 23.6%, and 16.4%, respectively).

The density of eels peaked at around 130–200 m sections where both large and small eels were present, and it decreased with increasing distance from the peaks where fewer mostly large eels were found (Fig. 6A). The density of small eels was consistently low at more reaches of the river upstream of the 300 m section (less than 0.02 eels $m^2$), especially considering that only 4 (August 2016), 0 (November 2016), 0 (July 2017), and 3 (September 2018) small eels were caught (Fig. S2). Eels were abundant just below the 310 m section, which was a pool habitat followed by concrete riverbank and sediment habitats at 320–350 m sections, and then the riverscape greatly changed to high elevation gradient (>10 m)

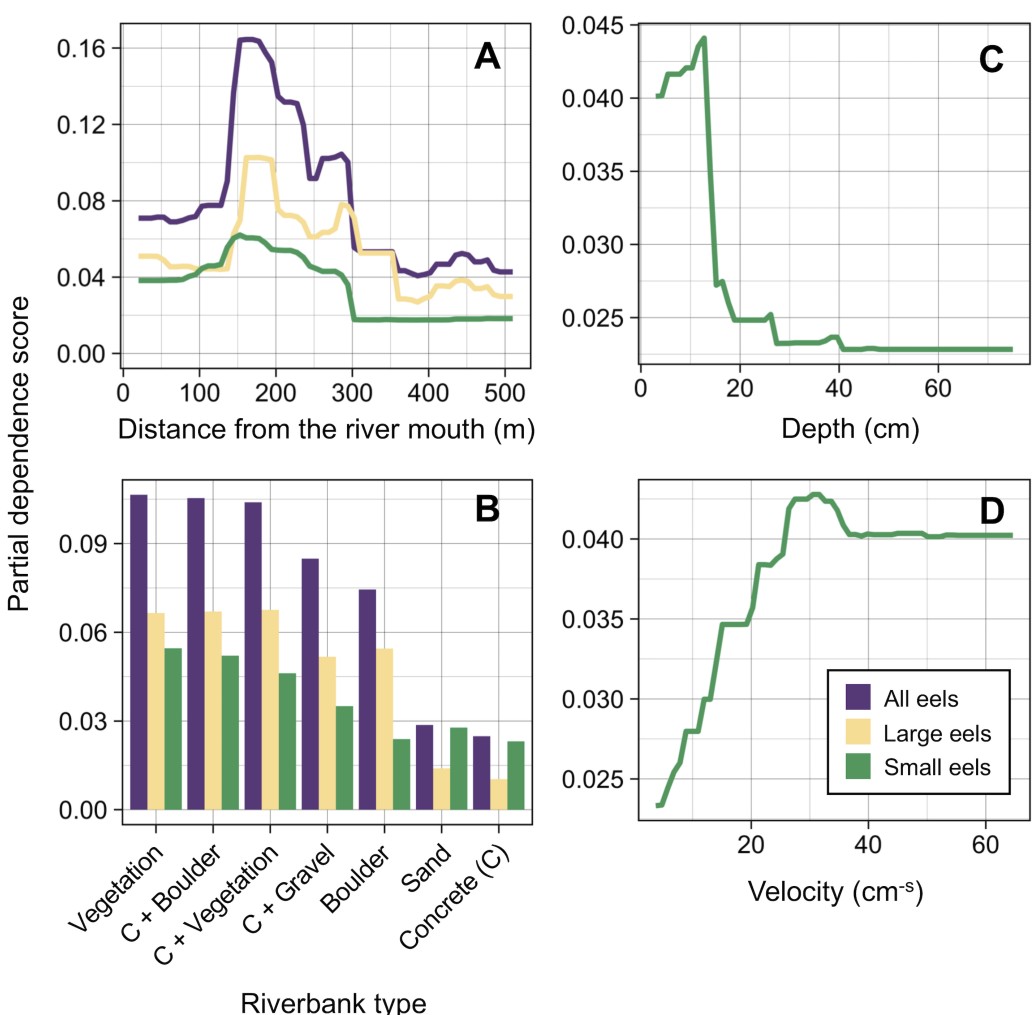

**Figure 6** **Modelled relationships of selected predictor variables by the random forest models for density of *A. marmorata* in the Oganeku River.** (A) Distance from the river mouth, (B) Riverbank type, (C) Depth (small eels), (D) Velocity (small eels). C, concrete.

upstream environments (boulder riverbank and boulder and gravel sediments) at the 360 m section (Figs. 2, 4). The density of eels was consistently lower among all three size-classes when riverbanks consisted of concrete and sand, while it was the highest when riverbanks consisted of vegetation, some of which extended into the water where some eels were collected (Figs. 6B and 7D). The density of small eels decreased when water depth was more than 15 cm (Fig. 6C) and when water velocity was less than 20 cm$^{-S}$ (Fig. 6D). In contrast, large individuals were found in a broader range of habitats with any depth and velocity (Figs. 7A and 7B).

Although sediment type was not selected the RF models for all three size-classes as a significant variable, the density of eels appeared to differ among sediment types (Fig. 6C). Eels were rarely found in sediment consisting of concrete and sand, and no large eels were found there. Almost all small eels were found in gravel sediment habitats that were

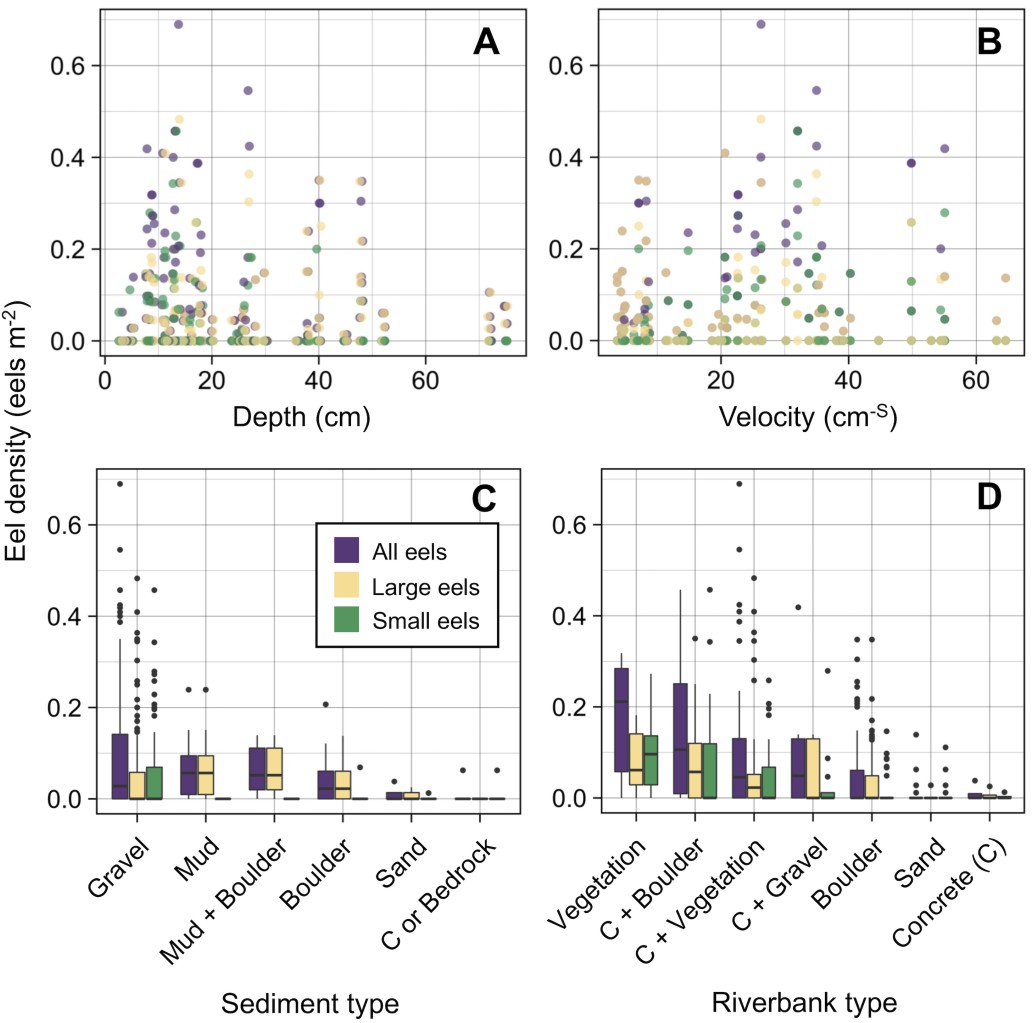

**Figure 7** **Density of *A. marmorata* for each environmental variable in the Oganeku River.** (A) Depth, (B) Velocity, (C) Sediment type, and (D) Riverbank type. In the boxplots, the middle lines indicate the median, the boxes represent the 0.25 and 0.75 quartiles, the whiskers are the values that are within 1.5 of the interquartile range, and the dots show outliers.

located from 70 m to 300 m sections (Figs. 4 and 7C), while they were rarely found in other sediment types. Large eels were found in a broader range sediment types with their densities being higher in mud and boulder sediments, and in combinations of habitats (Figs. 4 and 7C).

## Movements of tagged eels

Of the recaptured eels, 47.9% were recaptured from their original section of capture (i.e., distance travelled <10 m) (Figs. 8A and 8B), and the distance travelled of 75.9% of eels that were recaptured in a section that is different from the original capture section (i.e., distance travelled $\geq$ 10 m) were less than 50 m. Of the observed movement events ($n = 31$), 54.8% of the eels travelled in the upstream direction from the original section of capture (Fig. 8B).

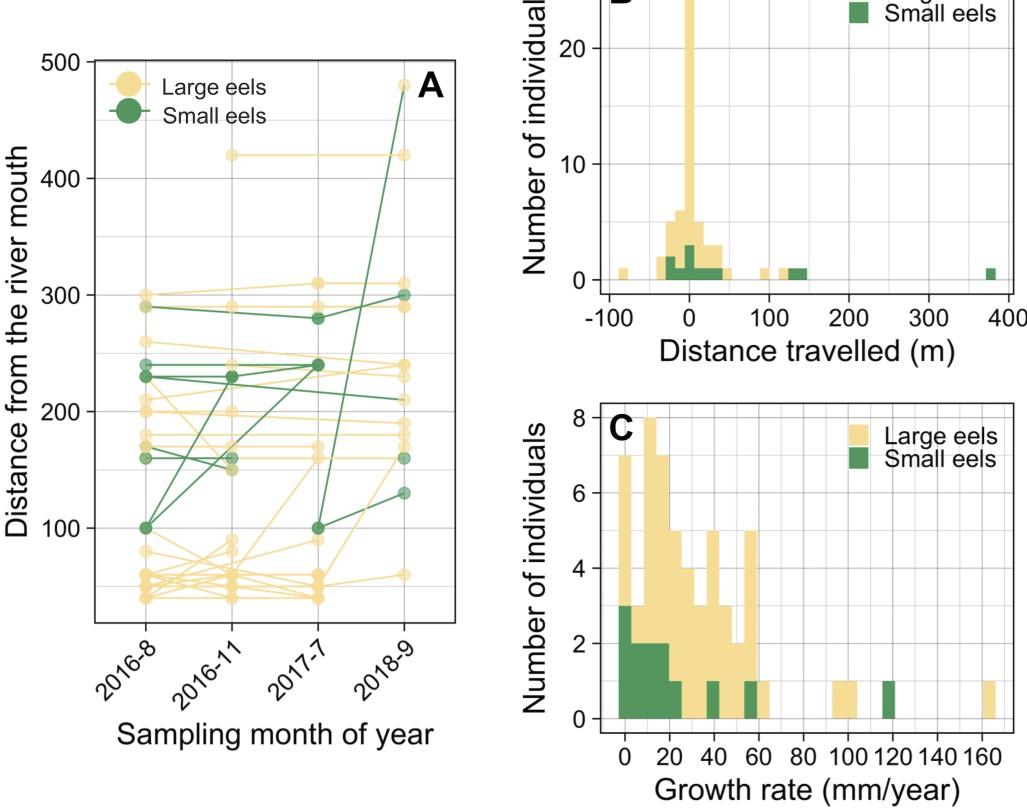

**Figure 8** **Movement and growth patterns of recaptured *A. marmorata* in the Oganeku River.** (A) Capture and recapture locations during each sampling survey connected by lines for each eel. (B) Histogram of distance travelled of the recaptured eels. (C) Histogram of the annual growth rates of the eels.

The distance travelled of recaptured eels that moved ≥ 10 m ranged from 10 m to 380 m with a mean ± SD of 46.5 ± 72.5 m (median = 20 m; $n = 31$). The distance travelled of recaptured large eels (TL ≥ 240 mm) ranged from 0 m to 120 m with a mean ± SD of 15.1 ± 26.5 m (median = 0 m; $n = 45$), while that of small eels (TL < 240 mm) ranged from 0 m to 380 m with a mean ± SD of 63.3 ± 110.8 m (median = 20 m; $n = 12$).

The proportion of movements more than 10 m was higher in small eels (75.0%) than in large eels (51.1%), but there was not a significant difference (Fisher's Exact Test, $P = 0.191$). The distance travelled of small eels (mean ± SD = 84.4 ± 121.9 m, median = 20) was greater than that of large eels (mean ± SD = 30.9 ± 31.0 m, median = 20), but there was not a significant difference (Exact Wilcoxon-Mann–Whitney test, $P = 0.262$; Fig. 8B). There were no significant relationships between the probability of occurrence of eel movement and TL at initial capture before each recapture (GLM: $P > 0.05$), irrespective of the possible distances at which the movement was observed (i.e., 10 and 20 m). The distance travelled of eels was significantly negatively related to TL at initial capture (GLM: coefficient ± SE = −0.003 ± 0.001, $t = −2.376$, $P = 0.018$), however, TL at initial capture

was not statistically different between eels that moved and those that did not move (Exact Wilcoxon-Mann–Whitney test, $P = 0.138$; Fig. S3A).

## Growth rate

The GR of recaptured eels ranged from 0 to 163.2 mm/year with a mean ± SD of 31.8 ± 31.0 mm/year (median = 24.1 mm/year; $n = 57$) (Fig. 8C). The best GLM having the lowest AIC value showed that the GR was significantly higher for eels that moved ≥80 m (GLM: coefficient ± SE = 1.798 ± 0.528, $t = 3.405$, $P = 0.001$) than for eels that moved <80 m (GLM: coefficient ± SE = 0.750 ± 0.320, $t = 2.344$, $P = 0.023$), both categories of eels that moved had significantly higher GR than the eels that did not move (Fig. S3B); although it is still unclear whether eels that moved ended up growing faster or faster growing individuals decided to move more. GR also increased with TL at initial capture (GLM: coefficient ± SE = 0.004 ± 0.001, $t = 3.856$, $P < 0.001$; Fig. S4A). The GR of eels was lower for small eels than that of large eels (Fig. S3C) and was different among the study periods (Fig. S4B), but these variables were not selected by the best model. Similarly, the sampling section distance from the river mouth was not selected by the best model for affecting GR. The eels that showed no growth between marking and recapture included five eels that were captured 3 months after tagging, and two eels that were captured after 11 and 25 months (Fig. S4A).

## DISCUSSION

### Habitat preference

The RF models revealed that the distance from the river mouth was consistently the most important variable to explain density of *A. marmorata* irrespective of size of eels. Few small eels were caught in the upstream sections > 300 m upstream and in sections < 100 m from the river mouth where large eels were present. Eels of all sizes were present from 100–300 m. As anguillid eels recruit from the sea to rivers, it has been well known for temperate eels that the density of eels in rivers is strongly related with the distance from the river mouth (*Smogor, Angermeier & Gaylord, 1995*; *Glova, Jellyman & Bonnett, 1998*; *Ibbotson et al., 2002*; *Yokouchi et al., 2008*; *Itakura et al., 2019*). For tropical eels, the abundance of *A. marmorata* was also significantly related with the distance from the river mouth when the species is highly dominant among anguillid species throughout rivers (*Robinet et al., 2007*; *Itakura et al., 2020b*; *Itakura et al., 2020a*), which is consistent with our findings. Therefore, the distance from the river mouth is likely one of the most common and important factors that determine the riverine distributions anguillid eel species.

The density of small *A. marmorata* was also negatively or positively related with water depth and velocity, respectively. Higher densities of small eels being present at shallower depths compared to in deeper areas was also found for temperate eels such as *A. japonica* (*Kume et al., 2019*), *A. anguilla* (*Laffaille et al., 2003*), *A. rostrata* (*Johnson & Nack, 2013*), and *A. australis* (*Glova, Jellyman & Bonnett, 1998*), and higher densities of small eels were also found in faster velocity waters for *A. anguilla* (*Laffaille et al., 2003*) and *A. dieffenbachii* (*Glova, Jellyman & Bonnett, 1998*). Small *A. marmorata*, therefore, seem to prefer shallow and fast-velocity waters (i.e., riffle or run, usually gravel or rocky sediments) rather than deep and slow-velocity areas (i.e., pools), although it should be noted that it is likely more

difficult to collect small eels using electrofishing in deep areas than shallow areas, which could partly cause the difference of catchability between these areas. The distribution of small *A. marmorata* was biased toward habitats that consisted of gravel sediment, possibly because all tidal limit areas in the study river where almost all small eels were collected consisted of the gravel sediment. The grain size of gravel as a refuge seems better for small eels than others, and so the riffle or run habitats with gravel may provide suitable refuges and feeding area for small eels. Conversely, such riffle and run habitats with gravel were also present in the upper reaches of the river, but the distribution of small eels was biased toward the lower reaches, suggesting that they prefer such habitats in the lower reaches. Because sediment was not selected by the RF models as an important variable, these results imply that the effect of sediment was masked by the strong effect of distance from the river mouth. As shown by this study and previous studies, the distance from the river mouth strongly contributes to determine eel density, which may mask microhabitat effects, especially when the survey was only conducted in a river that includes one pattern of distribution of habitat variables in relation to the distance from the river mouth. Therefore, further surveys in multiple rivers that have diverse distribution patterns of habitat variables are required to clarify microhabitat effects on eel distribution without the effect of the distance from the river mouth.

In contrast with small eels, the density of large *A. marmorata* was not significantly related to the microhabitat environments except for riverbank type. This indicates that the relationship between the microhabitat environments and eel density disappeared as the eels grow. Indeed, large individuals were found in broader habitats with any depth, velocity, and sediment type, and these factors were not selected by the final RF models. A similar finding was found for *A. japonica* in which the density of large eels ≥240 mm TL was not correlated with any depth, velocity, or sediments (*Japanese Ministry of Environment, 2016*). Such size-dependent changes in habitat use has also been reported for other anguillid species: large *A. dieffenbachii* are more uniformly spread across riffle, run, and pool habitats than small ones (*Glova, Jellyman & Bonnett, 1998*); *A. anguilla* progressively shift to deeper habitats as they grow (*Laffaille et al., 2003*); and small *A. japonica* used habitats near riverbanks, but large ones used habitats both near-riverbank and the center of rivers (*Kume et al., 2020*). Our results and those of previous studies suggest that eels appear to be able to flexibly use habitats having a variety of environments as they grow, which allow eels to move into and utilize the entire range of continental waters from saline bays, to entire river systems up to the headwaters if there are no obstacles (*Moriarty, 2003*).

Although *A. marmorata* seem to inhabit any habitat type as they grow, they appeared less likely to prefer habitats where the physical structure of the riverbank or riverbed was artificially altered by concrete. The RF models revealed that riverbank type also consistently contributed to explaining the density of eels as the second most important variable irrespective of size of eels. While the models estimated the highest densities to occur when riverbanks consisted of vegetation, riverbanks consisting of concrete and sand were estimated to have the lowest densities. Moreover, eels were rarely found in sediment areas that consisted of concrete and no large eels were found there. It was reported that abundance of *A. japonica* was lower in areas that consisted of concrete revetment compared

to those with vegetation or mud (*Itakura et al., 2015*), which corresponds to our findings. Such habitat modifications often result in reduced abundance and diversity of freshwater animals due to loss of structural diversity along riverbanks or riverbeds (*Taniguchi, Inoue & Kawaguchi, 2001*; *Wolter, 2001*). Thus, it is obvious that concrete does not provide suitable habitat for eels or their prey species because it is not possible to find shelter in concrete unless it is highly fractured, which may lead to the somewhat patchy distribution observed in this study. However, the density of eels was higher in sections that the riverbank consisted of concrete with boulders, vegetation, and gravel compared to that of concrete only. These combinations between concrete and the other materials made the density of eels almost identical to that in habitat consisting of vegetation. In addition, with regard to the combination between riverbank and sediment types, some eels were caught in the 110 and 120 m sections where the riverbank consisted of concrete with other types of sediments, while only one eel was caught in 320–350 m sections where both riverbank and sediment consisted of concrete. These results suggest that the combination-habitats of concrete and other materials may provide enough refuges for eels to inhabit the river at the current density levels.

However, habitat losses such as shoreline and riverbed modifications might cause higher densities of *A. marmorata* in the more suitable natural habitats by concentrating eels in those habitats. The resulting stronger intraspecific competition may lead to increased mortality or slower growth. A long life (mean ± SD = 12.8 ± 4.9 years; range = 3–30 on the study island: *Wakiya, Itakura & Kaifu, 2019*) with very slow annual growth, strong site fidelity, and size-dependent habitat preference of *A. marmorata* imply that they may be impacted by the habitat modifications of rivers on small islands such as the Oganeku River. Therefore, it is important to have long-term maintenance of diverse riverine habitats to conserve this eel species.

The TL of *A. marmorata* observed in the study river (336.8 ± 153.0 mm with a range of 62–770 mm) did not clearly differ with that in other rivers on the same island (385.5 ± 172.6 mm with a range of 119–1320 mm; *Wakiya, Itakura & Kaifu, 2019*); however, larger eels >800 mm in TL were absent in this study. It is well known that *A. marmorata* can frequently grow up to over a meter, but such larger eels are females only (*Hagihara et al., 2018b*). *Wakiya, Itakura & Kaifu (2019)* reported that males are dominant in other rivers in same island and there were very few larger female eels >800 mm. Although sex was not identified in this study, the majority of eels caught in the study river seems to be males that can start the spawning migration at <800 mm in TL, which may explain the absence of larger eels (females) there. The absence of the larger eels in the study river might be explained by the lack of much deeper pool waters that are preferred by larger eels, while rivers in *Wakiya, Itakura & Kaifu (2019)* have such deeper waters due to their larger river scale than the study river.

## Dispersal process after recruitment

The distributions of density and TL of *A. marmorata* provided interesting information about the dispersal process of the species after recruitment into the river. Eels that were <100 mm TL were mostly found in the sections 100–200 m from the river mouth where

the freshwater tidal limit was located (100–150 m), while small eels <240 mm TL were found at a wider range of sections between 100–300 m. The GAM model showed that the TL of eels had a minimum peak at around the tidal limit of the 100–150 m sections, and increased with increasing distance from the tidal limit. Moreover, the RF models revealed that densities of all eels, small eels, and large eels peaked at around the tidal limit, and they decreased with increasing distance from the tidal limit. These density and size gradients of eels in relation to the distance from the river mouth indicate that *A. marmorata* initially recruited to freshwater tidal limit areas after recruitment into the river and then dispersed in both downstream (<75 m) and upstream directions as they grew. No glass eels and few smaller eels of *A. marmorata* were collected in this study, partly because it is difficult to capture them using electrofishing. Nevertheless, we collected small eels less than 100 mm TL with the smallest individual being 62 mm TL, which would likely be individuals that recruited to the river within the last year. Therefore, *A. marmorata* arriving from the sea into the river seem to initially settle in the tidal limit area.

This type of dispersal process after recruitment may be common among anguillid eels because similar processes have been reported for temperate eels such as *A. anguilla* (*Edeline et al., 2007*; *Costa et al., 2008*) and *A. japonica* (*Kaifu et al., 2010*; *Wakiya et al., 2019*). Glass eels and the subsequent smaller growth-phase eels are often abundant at the freshwater areas of the upper estuaries (tidal limit) of rivers and their sizes increase with increasing distance from the tidal limit, while their densities peaked at the tidal limit (*Haro & Krueger, 1991*; *Daverat & Tomás, 2006*; *Aprahamian et al., 2007*; *Edeline et al., 2007*; *Costa et al., 2008*; *Kaifu et al., 2010*; *Wakiya et al., 2019*). Conversely, a more homogenous distribution of smaller growth-phase eels was found in an estuary compared to the glass eels (*Edeline et al., 2007*). These size and density gradients of eels are thought to result from the dispersal behavior of eels, which shows that glass eel arriving from the sea accumulate at the freshwater tidal limit of an estuary and then disperse in both more downstream and upstream directions as they grow. Our first of their kind results for tropical eels are in agreement with these previous studies. The common point between *A. marmorata* in this study and these temperate eel species is that each eel species is highly dominant among anguillid species throughout the rivers in each distributional area. In our study river, only 7 Japanese eels were caught during the 4 sampling periods, so they are clearly not abundant compared to *A. marmorata*, or most do not survive very long. The dispersal process of *A. marmorata* shown in this study may change depending on whether sympatries of multiple eel species occur within same watersheds (*Marquet & Galzin, 1991*; *Shiao et al., 2003*; *Arai & Abdul Kadir, 2017*; *Hagihara et al., 2018a*; *Hsu, Chen & Han, 2019*). Therefore, future research conducted in regions with sympatries of multiple eel species will help to further understand how these sympatric eel species disperse to each habitat in river systems.

This dispersal behavior might be an adaptive strategy to increase individual fitness by reducing intraspecific competition. The mortality of eels in freshwater may primarily be related to density-dependent factors such as intraspecific competition for resources or predation by eels (*Vøllestad & Jonsson, 1988*). Because higher density would lead to stronger intraspecific competition, the eel movements from the tidal limit to both downstream and upstream directions can be regarded as density-dependent dispersal that mitigates the
competition (*Edeline et al., 2007*; *Kaifu et al., 2010*). Homogenous distribution of smaller growth-phase eels around the tidal limit also suggests density-dependent dispersal (*Edeline et al., 2007*). As shown in this study, the growth-phase eels after dispersal and subsequent settlement in habitats exhibit strong sedentary behavior and limited movements (*Jellyman & Sykes, 2003*; *Ovidio et al., 2013*; *Itakura et al., 2018*). Accordingly, the dispersal-related movements may be one of the key elements to mitigate the intra-interspecific competitions during their growth stage.

## Movements of eels

The mean distance travelled of the tagged large *A. marmorata* was $15.1 \pm 26.5$ m (median = 0 m), which is consistent with previous studies for temperate eels showing that most growth-phase anguillid eels have limited movements (*Gunning & Shoop, 1962*; *Parker, 1995*; *Jellyman & Sykes, 2003*; *Ovidio et al., 2013*; *Itakura et al., 2018*). Half of the eels were recaptured from the original section of capture (i.e., distance travelled <10 m), suggesting strong fidelity of growth-phase *A. marmorata* to 'familiar' habitats, as mentioned for *A. japonica* (*Itakura et al., 2018*). Furthermore, the distance travelled of small eels was greater than that of large eels, which is likely related to the upstream or downstream dispersal behavior after recruitment as discussed above, while large eels show sedentary behavior after they establish a home range (*Imbert et al., 2010*; *Wakiya, Kaifu & Mochioka, 2016*).

Because we performed the sampling during daytime only, the short distance travelled reported here was limited to their movement between daytime refuges. Considering that anguillid eels are generally nocturnal (*Parker, 1995*; *Jellyman & Sykes, 2003*; *Ovidio et al., 2013*; *Itakura et al., 2018*), it is likely that growth-phase *A. marmorata* move longer distances during night than observations in this study have indicated. In addition, the sampling events were conducted only four times over the 2-year study period, which make it difficult to estimate more exact distances travelled of eels or the sizes of their home ranges. Therefore, a mark-recapture experiment with more intensive sampling during both daytime and night or more continuous studies using methods such as biotelemetry are needed to further understand the movement ecology of this species.

## Growth

The direct measurement of individual growth by the mark-recapture experiment revealed that the *A. marmorata* in this river had a very wide range of annual GR among the 48 recaptured eels that ranged from 0 to 163.2 mm/year, with a mean $\pm$ SD of $31.8 \pm 31.0$ mm/year. This mean GR value is generally consistent with the otolith-based estimate value ($25.9 \pm 6.6$ mm/year) obtained from other rivers on same island (*Wakiya, Itakura & Kaifu, 2019*). This indicates that otolith-based age estimates of this species in the previous study seem to be reasonable and that the otolith analysis method can be useful for the estimating age of *A. marmorata*. The range of GR from this study was much wider than the otolith-based estimate values (15.8–50.2 mm/year) from *Wakiya, Itakura & Kaifu (2019)*. Surprisingly, some individuals showed no (zero) or very high (>100 mm/year) annual growth. Compared to our results, the mean GR of *A. marmorata* in an equatorial region (Sulawesi Island Indonesia) was three times higher than that in this study (*Hagihara et*

al., 2018b). As discussed by *Wakiya, Itakura & Kaifu (2019)*, the differences in GR of the species among latitudinally different regions may partly be explained the differences in annual water temperatures and productivity in the growth habitats. These results suggest that *A. marmorata* may accommodate any growth situations including extreme low and high growth conditions in response to a variety of environments. This diverse growth pattern of the species might allow eels to adapt to various habitats in continental waters in latitudinally expanded distributional regions from the equator to higher latitude regions such as southern Japan (*Mizuno & Nagasawa, 2010*).

In our study that extended across 2 years, there might have been seasonal and interannual differences in temperature and food availability that could have affected the GR of eels during each marked and recapture period that were of various durations, although study period was not selected by the best model. Temperature is one of the main seasonal and interannual effects on eel growth (*Daverat et al., 2012*; *Yokouchi & Daverat, 2013*), but this may typically also be linked to seasonal cycles of prey availability. Eels that were caught in July 2017 and recaptured in September 2018 would have experienced an entire seasonal cycle in the river, and they had the highest GR among eels in this study. Conversely, eels that were caught in August 2016 and recaptured in November 2016 only experienced three months that did not include the spring season, and those eels had the lowest median GR and the highest range of GR values. Further research is required to examine the seasonal differences of growth of *A. marmorata* by conducting a mark-recapture experiment with seasonal intervals.

Another potential reason to explain the difference in GR of eels among the study periods is possible seasonal differences in food availability for the eels. Although we did not document the seasonal patterns of their species composition and abundance, a total of 33 diadromous fish and crustacean species was found in the study river during the sampling surveys, which are likely potential prey species for eels. In other rivers of the study island, the biomass of these fish and crustacean species accounted for more than 80% of stomach contents of *A. marmorata* (R Wakiya, 2015, unpublished data). These diadromous species can have seasonal patterns of recruitment into rivers with their own phenology (*Tanaka et al., 2020*), so their recruitment dynamics might lead to seasonal and interannual differences in food availability in the river, which could affect the GR of eels. A greater diversity of fish and crustacean species and higher overall abundance appeared to be present in the lower river reaches below about 320 m from the river mouth (Table S1), which might be one reason why few eels were found in the more narrow upper reaches that likely have a lower carrying capacity for eels.

## CONCLUSIONS

Our study is the first to provide information about several aspects of the riverine ecology of the spatial distribution, growth, and movement of the tropical eel *A. marmorata* in relation to environmental conditions, because we conducted a mark-recapture experiment across 2 years throughout the main reaches of the Oganeku River on Amami-Oshima Island, Japan, where it was the highly dominant anguillid species compared to small numbers of Japanese

eels. This ecological information about *A. marmorata* in a small subtropical island river can be compared to future studies in different regions and will contribute to conservation and management efforts for anguillid eels in the Indo-Pacific.

## ACKNOWLEDGEMENTS

We are deeply grateful to K Ebihara, M Gollock, K Iwabuchi, K Kaifu, M Matsuoka, M Sakai for their help in the field sampling.

### Funding
Hikaru Itakura was financially supported by a Research Fellowship for Young Scientists and a Postdoctoral Fellowship for Research Abroad from the Japan Society for Promotion of Science. This study was supported by the River Fund of the River Foundation, Japan, the Sasakawa Scientific Research Grant from the Japan Science Society, and the Environmental Research Fund of the Ministry of the Environment, Japan. The funders had no role in study design, data collection and analysis, decision to publish, or preparation of the manuscript.

### Grant Disclosures
The following grant information was disclosed by the authors:
Research Fellowship for Young Scientists, from the Japan Society for Promotion of Science.
Postdoctoral Fellowship for Research Abroad from the Japan Society for Promotion of Science.
River Fund of the River Foundation, Japan.
Sasakawa Scientific Research Grant from the Japan Science Society.
Environmental Research Fund of the Ministry of the Environment, Japan.

### Competing Interests
The authors declare there are no competing interests.

### Author Contributions
- Hikaru Itakura conceived and designed the experiments, performed the experiments, analyzed the data, prepared figures and/or tables, authored or reviewed drafts of the paper, and approved the final draft.
- Ryoshiro Wakiya conceived and designed the experiments, performed the experiments, authored or reviewed drafts of the paper, and approved the final draft.

### Animal Ethics
The following information was supplied relating to ethical approvals (i.e., approving body and any reference numbers):

Kagoshima Prefecture approved the study (approval number: 2006-5 for 2016 and 2006-10 for 2017 and 2018).

## Data Availability

The raw measurements are provided in the Supplemental Files.

## Supplemental Information

Supplemental information for this article can be found online at http://dx.doi.org/10.7717/peerj.10187#supplemental-information.

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
