# Peer review of "Habitat preference, movements and growth of giant mottled eels, Anguilla marmorata, in a small subtropical Amami-Oshima Island river"

_PeerJ, doi:10.7717/peerj.10187_

## Round 0.1 · original submission · Minor Revisions

The reviewers have identified a number of issues that need to be addressed. Please indicate how you have addressed each reviewer comment in your reply. In addition to their comments, please consider how your statistical model confounds ‘distance from river mouth’ and environmental variables and whether there is an approach that could identify independent effects.

Some specific comments follow:
Line 131. Replace ‘throughout each river’ as ‘throughout the river network’
Line 142. Replace ‘have’ with ‘has’
Line 170. Replace ‘greatly changed’ to ‘transitioned’
Line 171. Delete ‘having a’
Line 196. Replace ‘wooded’ with ‘wooden’
Line 314 and elsewhere. Avoid use of ‘respectively’. Reword to put number of individuals in parentheses after each year. Makes less work for the reader.
Line 382. Delete ‘that’
Line 389. Replace ‘regarded’ with ‘observed’
Lines 406-408. Sentence is not necessary. Consider deleting.
Lines 405-423. This paragraph uses many words to describe activities of a few eels. Consider condensing to key points.
Line 427. To what extent is distance from the river mouth confounded with habitat variables?
Line 438. Seemed like concrete was very important. Refer to previous comment.
Line 450. Similar issue as previous two comments. Need a statistical method that does not confound environmental variables with distance to river mouth. Consider separate analyses for distance to river mouth and the habitat variables.
Lines 480-489. This part is especially confusing. Consider rewriting to simplify.
Lines 550-560. This paragraph adds little to the discussion. Consider deleting.
Line 568. Replace ‘seems with ‘seem’
Lines 582-591. Need to break growth rate estimates into seasonal components.
Conclusion. This adds little new information. Consider deleting.

Reviewer 1 ·

Basic reporting

This paper describes a 2-year mark-recapture study of juvenile and subadult Anguilla marmorata in a relatively small stream in southern Japan. The focus of the work is on distribution, preferred habitat, home range, and age and growth. The findings add to a growing body of ecological knowledge for this and other tropical species of Anguilla, which up to now have been little studied.

The manuscript is relatively well researched and draws on an extensive list of references. Most references are more recent (post-2000); some older but relevant references on other species of Anguilla might be included, e.g. Gunning and Shoop 1962, Haro & Krueger 1991 (see citations in General Comments). The text contains few technical, grammatical, and stylistic errors, and I have relatively few comments/edits. The authors sufficiently review recent research with this and other subtropical Anguilla species. To my knowledge this research has not been published elsewhere.

Experimental design

The experimental design follows that for a standard mark-recapture study. Tag retention and fish catchability are not quantified, but the tagging methods have been used with juvenile Anguilla before and are probably appropriate. Some of the statistics employed are new to me (e.g. random forest models) but seem appropriate given limitations on sample size, non-normality of data, etc. The authors do a good job of justifying the specific analyses used.

Catchability of eels using backpack electrofishing may be influenced by habitat (i.e., it is generally harder to detect and catch small eels in deep runs or areas with cover with electrofishing), which may have affected results somewhat (i.e., fewer small eels collected in deeper areas). However, there are no data to evaluate this potential effect. The recapture rate of eels in this small system, which seems to be easily comprehensively sampled, is quite small (48 of 339 fish = 14% recap rate; line 322). This might imply somewhat limited catchability using the electrofishing technique. Can the authors speak to this?

Validity of the findings

The study was performed in a lower portion of a relatively small river system only 500 m long; how representative as a general habitat is this for this species? I would imagine larger/longer river systems might have different size/age distribution characteristics. Can the authors make any comparisons between this and other riverine habitats for this species?

Eel density ranged from 0 to almost 0.7 eels/m2. How does this compare to other studies of density of tropical and temperate eels?

My understanding is that adults of this species can grow to a very large size (2+ m total length), but few eels over 600 mm TL were collected. Why were larger/older eels absent from the system? Could the stream size have been too small to support them, or do larger eels prefer other habitat types (e.g., larger rivers)?

The authors mention the potential for competition and habitat segregation of sizes in this river system (lines 436, 528). Is there any additional evidence for this? Are mottled eels the dominant fish species (in terms of number or biomass) in this system? If so, one might also look for evidence of cannibalism (which is common in Anguilla); the authors note that no glass eels and very few very small eels were collected throughout the 2 years of study.

Along these same lines, in temperate eels large runs of glass eels usually invade streams in the spring months and attempt to migrate as a far upstream as possible in their first year. If this is the case for mottled eels, one might expect large concentrations of glass eels accumulating below the waterfall 520 m from the river mouth (albeit perhaps during a very limited period of the year). Can small eels ascend this waterfall? If so, is there suitable habitat above the waterfall? If they cannot ascend, could they be consumed by larger eels or other predators? The authors make the claim that juvenile eels enter the river and establish themselves primarily in the riffle region 100-300 m above the river mouth but have no direct evidence of this. Because they were rarely observed, alternative routes and/or fates of glass eels entering from the ocean into this small stream should also be acknowledged.

Additional comments

Title: capitalize “island” if Amami-Oshima is indeed a specific named island

Line 38: change “main reaches” to “mainstem reach”

Line 57: change “is” to “was”

Line 86: change “most species have not been studied for their freshwater ecology” to "their freshwater ecology has not been studied"

Line 131: “…A. marmorata is highly dominant…” - does this mean dominant among other Anguilla species or among other fishes in general?

Line 160: provide latitude and longitude of study site

Line 169: sentence not clear; reword

Line 171: state (if known) whether the waterfall is passable by eels

Line 173: omit “to”

Line 173: how long is the total length of the mainstem, including the portion above the waterfall?

Line 179: change “was” to “were”

Line 183: “entire river” – there was no sampling above the waterfall; reword

Line 190: “uppermost reaches - there was no sampling above the waterfall; reword

Line 192: omit "of every sampling time"

Line 201: Few A. japonica were collected; does this imply habitat segregation of the two species? A. japonica is infrequently mentioned in the Discussion; habitat segregation (or not) of the two species could be discussed more there.

Line 233: change “detected” to “quantified”

Line 367: change “was” to “were”

Line 396: I assume the remaining sentences in this paragraph refer to the three eels that made the longest movements.

Line 401: omit comma

Discussion: The Discussion section is rather long, but still interesting. The authors speculate a bit, but perhaps are warranted in this given how little is known for this species. Editor’s call.

Line 467: change “range” to “range of”

Line 551: A home range of 15.1 +/- 26.5 m is pretty small. The collections were made during daytime only, perhaps collecting eels only while in they were in their daytime refuges, which they may have had some fidelity to. Movements a night (e.g. ranging much farther from a daytime refuge) may be much more extensive; the authors should acknowledge this.

Fig. 5; why are no large eel results shown in Figs. 6C and 6D?

Figures: In the interest of visualizing the data, it would be nice to include in the main manuscript a figure for density vs. distance, for both small and large eels, perhaps also showing sediment and bank type (i.e., Fig. S2). I would suggest breaking up Fig 3 and retain Fig.3A as a stand-alone figure, then add a new figure that emulates Fig. S2, adding the predictive value and 95% CI to the rightmost panel of S2. The open circles for A. japonica can be omitted.

Suggested references:

Gunning, G. E., and C. R. Shoop. 1962. Restricted movements of the American Eel, Anguilla rostrata (LeSueur) in freshwater streams, with comments on growth rate. Tulane Studies in Zoology 9:265-272
.
Haro, A. J., and W. H. Krueger. 1991. Pigmentation, otolith rings, and upstream migration of juvenile American Eels (Anguilla rostrata) in a coastal Rhode Island stream. Canadian Journal of Zoology 69:812-814.

Reviewer 2 ·

Basic reporting

The manuscript is readable, and the study appears to be conducted by experts on the subject matter.

Experimental design

Methods are well described and statistical analyses appropriate. I have several major issues with the Methods section.

(1) Growth: Given the irregular sampling intervals that range 3-14 months, it is challenging to interpret growth rate data. Authors wisely acknowledge the issue on line 582-591, and relegate data to Appendix Figure S5. On Figure S5, it seems growth rates appear to be higher when intervals contain spring, which I suspect may be the season with the highest growth rate. This is a major, if not fatal, shortcoming, which at this point is hard to offer a constructive suggestion.

(2) Home range: I don't think that this phrase is used appropriately in this manuscript, as estimating home range requires location data at finer temporal scales (i.e., kernel density estimates). The phrase in fact is not defined clearly in the manuscript, but it is referred to as "long-term linear home rage", which I take to mean cumulative distances for individuals recaptured more than once. Four recapture events over > 2 years are not simply sufficient for home range estimation, and I advise that authors refrain from using this phrase throughout.

(3) Movement: Authors conclude that movement is limited because 47.9% were recaptured in the same 10m section, but I wonder if the short study area (450m) provides a classic example of missing emigration from the area. The finding that smaller individuals traveled longer distances than larger individuals (Line 384-385) could just a symptom of the issue, where larger, mature individuals moved to the ocean. This would result in detecting only those that stayed in the study area (underestimating long-range movement). This potential criticism is further substantiated by the fact only 48 out of 396 individuals were recaptured, although it is possible that low recapture rates are due to other factors such as morality and detection rates by electrofishing. I suggest that authors discuss the potential impacts of the study design on underestimating movement, and even better, demonstrating why this would not be an issue in this study.

(4) Electrofishing: I wonder whether electrofishing was effective at capturing animals that burrow into substrates in brackish waters. I have a mixed experience with electrofishing when water conductivity is high, and species like juvenile eels and lamprey ammocetes appear quite elusive. It is particularly problematic when electrofishing capture probability varies by habitat. For example, on Figure 6(A), eel density decreases with depth. Is this because eels are harder to catch at deeper habitats? I suggest authors demonstrate that electrofishing capture probability is not habitat dependent (ideally) or at least report the range of conductivity along the study river.

Validity of the findings

The key take-home message of this manuscript is: "Given the long life, slow growth, strong site fidelity, and size-dependent habitat preference of A. mamorate, the long-term maintenance of diverse natural riverine habitats are important to conserve this eel species (Line 64-66)." This blanket statement (sounds like a cliche) is likely right, but when examining data presented in the paper carefully, I wonder if the statement is sufficiently back up by data.
- long life: true, but not a focus in this paper (no data presented).
- slow growth: yes, there were a couple of individuals that did not grow at all for months, but there were others that grew much faster. What is striking seems growth variation among individuals, not necessarily slow growth.
- strong site fidelity: this is questionable (see Comment 3 above)
- size-dependent habitat preference: Given that large eels were habitat generalists, this applies more to small eels that occupied riffles/runs.

I wonder whether an alternative take-home message, supported more strongly by data in this paper, would be appropriate. For example, authors emphasize many similarities of the giant mottled eel to better-studied temperate eels. Would you be able to say that many conservation strategies that apply to temperate eels apply similarly to tropical eels, etc?

Additional comments

A few other, more minor comments:

- line 211: The total sample reported in Abstract (line 41) and Results (line 312) is 396 individuals. But here, 228 PIT tagged individuals + 105 VIE tagged individuals = 333 individuals?

- line 219: Was there only a single measurement of depth and velocity per 10m section? This should have been taken at 3 or so points across multiple transects per section. At the least, you are measuring meso-habitat characteristics, instead of micro-habitat characteristics, because you cannot exactly know where eels were within a 10m section.

- line 408-412: It is challenging to discern in this study (and many others) whether individuals that moved ended up growing faster, or faster growing individuals decided to move to different locations.

---

## Round 0.2 · accepted · Accept

Thank you very much for your thoughtful and thorough revision. You did an excellent job addressing and responding to all comments.